# Transurethral Resection of Non-Muscle Invasive Bladder Tumors Combined with Fluorescence Diagnosis and Photodynamic Therapy with Chlorin e_6_-Type Photosensitizers

**DOI:** 10.3390/jcm11010233

**Published:** 2021-12-31

**Authors:** Andrey V. Kustov, Nataliya L. Smirnova, Oleg A. Privalov, Tatyana M. Moryganova, Alexander I. Strelnikov, Philipp K. Morshnev, Oscar I. Koifman, Alex V. Lyubimtsev, Tatyana V. Kustova, Dmitry B. Berezin

**Affiliations:** 1United Physicochemical Centre of Solutions, G.A. Krestov Institute of Solution Chemistry, Russian Academy of Sciences (ISC RAS), 153045 Ivanovo, Russia; smir141973@mail.ru (N.L.S.); morshnevphilipp@gmail.com (P.K.M.); 2Ivanovo Regional Clinical Hospital (IRCH), 153000 Ivanovo, Russia; privalovdoktor@mail.ru (O.A.P.); tanya42229@mail.ru (T.M.M.); strelnikovprof@gmail.com (A.I.S.); 3Department of Faculty Surgery and Urology, Ivanovo State Medical Academy (ISMA), 153012 Ivanovo, Russia; 4Institute of Macroheterocyclic Compounds, Ivanovo State University of Chemistry and Technology (ISUCT), 153012 Ivanovo, Russia; oik@isuct.ru (O.I.K.); alexlyubimtsev@mail.ru (A.V.L.); melenchuktv@mail.ru (T.V.K.); berezin1@rambler.ru (D.B.B.)

**Keywords:** bladder tumors, transurethral resection, high recurrence, fluorescence diagnosis, photodynamic therapy, chlorin photosensitizers

## Abstract

Bladder cancer is a common disease with a high recurrence rate. In order to improve the treatment of superficial bladder tumors, we evaluated the efficacy and safety of transurethral resection (TURB) followed by fluorescence diagnosis (FD) and photodynamic therapy (PDT) with chlorin e_6_ photosensitizers (PSs), *viz*. “Fotoran e_6_” and “Fotoditazin”. It was found that both PSs generated singlet oxygen and revealed moderate affinity toward the lipid-like compartment. Between November 2018 and October 2020, 12 patients with verified non-muscle invasive bladder cancer (NMIBC) were treated by TURB combined with FD and PDT. Eight patients received “Fotoran e_6_” intravenously, while four patients received intravesical PSs. The patient ages were between 31 and 79 years, with a median age of 64.5 years (mean 61.3 ± 14.2). The total light dose was 150 J/cm^2^ for the local irradiation of the tumor bed with a red light at the *λ* = 660 nm wavelength, and 10–25 J/cm^2^ were additionally delivered for diffuse irradiation of the entire bladder mucosa. At the median follow-up period of 24 months (mean 24.5 ± 5.4 months, range 16–35 months), 11 patients remained tumor-free. One 79-year-old patient developed a recurrence without progression to the muscle layer. This pilot study shows that the TURB + FD + PDT technique is an effective and safe option for the first-line treatment of superficial bladder tumors.

## 1. Introduction

Bladder cancer (BC) is known to be the eleventh most commonly diagnosed malignancy worldwide [1,2,3], affecting men more frequently than women. BC incidence and mortality rates vary from country to country depending on differences in risk factors and diagnostic practice [2,3]. In the Russian Federation, the incidence rate of bladder cancer is reported to be within 2–5%, and these malignancies are ranked number two among the oncological diseases of the urinary tract [4]. TURB in white light, followed by intravesical chemotherapy (CT) or bacillus Calmette–Guérin instillations (BCG), is considered to be the gold standard for treating NMIBC all around the world [3,4]. However, satellite malignancies are often invisible in white light, which may lead to early recurrence. Fluorescence-guided biopsy and narrow-band imaging are much more sensitive than conventional procedures in detecting superficial tumors, especially for flat, high-grade malignancies (CIS) [3]. Despite a number of contradictive results, meta-analysis has recently confirmed that BCG instillations after TURB are superior to TURB alone or TURB followed by intravesical chemotherapy in preventing tumor recurrence [3]. However, the BCG procedure is unavailable in most regional clinics and is often associated with serious side effects affecting up to 5% of patients [2,3]. The recurrence rate for bladder tumors during the first year after TURB + CT is estimated to be within 36–44%, while TURB + BCG reduces this value to 20% (see [4] and references therein). 

The TURB + FD + PDT treatment with photosensitizers of the second generation is considered a safe and efficient alternative to TURB + BSG or TURB + CT, especially among patients with recurrent bladder cancer and patients with malignancies that are resistant to BCG or chemotherapy [4,5,6,7,8]. Photodynamic therapy proved to be a successful therapeutic modality for the management of many neoplastic and non-malignant diseases, including chronic infections of the urinary tract [5,9,10,11,12,13,14,15,16,17,18,19,20]. This easily repeatable procedure is a well-established tool for inactivating Gram-positive and Gram-negative bacteria [9,10,14,15], bacterial biofilms [12], fungi [9,13], and viruses [9,11], and it is widely used in oncology to treat superficially located tumors [16,17,18,19,20]. PDT consists of three essential non-toxic components: a photosensitizer, light, and molecular oxygen [5,9,17]. All three components are non-toxic, but together they initiate a cascade of photochemical reactions that lead to the generation of reactive oxygen species (ROS)—the most important of them being singlet oxygen, ^1^O_2_ [17]. The latter has a very short lifetime of several microseconds or even smaller [17,21,22,23,24], which limits its diffusion in living cells and restricts the damaged area. However, ^1^O_2_ rapidly induces significant toxicity, leading to cell death via apoptosis, necrosis, or autophagy [5,17,21,24]. In general, antitumor effects of PDT consist of three inter-related mechanisms: a direct cytotoxic effect on tumor cells, damage to the tumor vasculature, and the induction of a robust inflammatory reaction leading to the development of systemic immunity [9,17]. It is believed that PDT may activate both humoral and cell-mediated antitumor immunity, although the importance of the former response is less clear. 

Most of the photoactive molecules in PDT have a tetrapyrrole structure similar to that in natural dyes, such as hemoglobin or chlorophyll. These species are capable of being locally or systemically administered into a malignant tissue [5,9,17]. The interval between PS administration and irradiation is rather long, and the sensitizer molecules are given enough time to diffuse from normal tissues. In contrast, the PS molecules localized in malignant tissue remain there. Hence, they are able to absorb a photon during irradiation and transfer its energy to the surrounding oxygen molecules to initiate the cell death mechanisms mentioned above [17,21,22,23,24]. 

The first-generation PSs, based on a porphyrin platform, had long-lasting skin photosensitivity and relatively low absorbance in red light [9,17]. The application of these drugs for treating superficial bladder tumors was, however, quite successful and provided the early complete response rate of 45–84% in patients with recurrent bladder tumors [9,17]. The second-generation PSs, such as 5-aminolevulinic acid (ALA), its esters, or several chlorin e_6_ derivatives, were found to be more efficient and safe [4,5,6,7,8]. In particular, the TURB + PDT treatment with ALA and “Radachlorin” provided 78% and 91% complete response rates, respectively, at 12 months after follow-up [4,8]. However, by two years, the recurrence rate achieved for the “Radachlorin” group was 35% [8,17]. Similar results were obtained in a small cohort of five patients with recurrent bladder cancers treated with the chlorin PS “Fotolon” [7]. 

With these results in mind, we have initiated this pilot single-center study of the TURB + FD + PDT efficacy with the second-generation chlorin PSs, *viz*. “Fotoditazin”, and “Fotoran e_6_”, (see Figure 1) in 12 patients with superficial T1 bladder tumors, in the hope that this modality provides a disease-free period comparable to that after TURB + BCG standard therapy. Additionally, some crucial physicochemical parameters for both PSs were determined and briefly discussed.

## 2. Materials and Methods

### 2.1. Study Description and Patient Selection Criteria

This single-center pilot study—was performed in a small cohort of 12 patients between November 2018 and October 2020 in agreement with applicable laws and regulations, clinical practices, and ethical principles described in the Declaration of Helsinki. The informed consent was obtained from each human subject and kept in the urological clinic of IRCH. FD and PDT options were approved in each case by the decision of the Medical Commission of IRCH for patient selection for high-tech medical care (protocols 2018–2020, according to the Appendix to the order of the Chief Physician, No. 225, dated 20 July 2015) and also by the decision of the Ethical Committee of ISMA (protocol No. 8, dated 6 October 2021). Twelve patients with the histologically verified T1 NMIBC were recruited from the urological clinic of the Ivanovo regional hospital. All of them had had no prior BCG treatment. According to the WHO grading [3,4], 8 patients had well-differentiated tumors (G1), whereas moderately differentiated tumors (G2) were diagnosed in 4 patients. The exclusion criteria were identical to those described elsewhere [8] and included muscle-invasive disease (≥T2), porphyria, maximal bladder capacity < 200 mL, gross hematuria, psychiatric disorder, etc. 

### 2.2. Photosensitizers and Other Chemicals

“Fotoditazin” was purchased from the “VETA-GRAND LLC” company (Moscow, Russia) as a 5 mg/mL concentrate. This solution was stable in the dark at 283 K for one year. “Fotoran e_6_” was provided by the “RANFARMA” company (Krasnogorsk, Russia) as a solid powder mixed with polyvinylpyrrolidone as an appropriate carrier. Both PSs were also purchased in their pure solid state (final purity > 93%). 1-Octanol (Panreac, Barcelona, Spain, >98%) was dried with 4 Å molecular sieves and distilled under reduced pressure at 355 K. The phosphate saline buffer (PSB) pH = 7.4 (Agat-med, Moscow, Russia, for biochemical laboratories) was prepared by dissolving pure solid forms in one liter of freshly bidistilled water [14,15]. 1,3-diphenylisobenzofuran (DPBF) (J&K Scientific Gmbh, Stadtkreis, Germany, >97%), as a selective singlet oxygen quencher, was used as supplied.

### 2.3. Physicochemical Studies 

UV-Vis spectra were recorded with a D8 spectrophotometer (Drawell, Chongqing, China), both in 1-octanol (OctOH) and water, at 293 K. Fluorescence spectra were registered with a CM 2203 spectrofluorimeter (Solar, Minsk, Belarus). 

The singlet oxygen quantum yield (*Φ*_Δ_) was determined using an indirect chemical method, which was similar to the technique described elsewhere [25,26]. Briefly, two quartz cuvettes containing a solution of PS + DPBF in OctOH and a solution of standard PS + DPBF were irradiated with an LED panel [14,15]. The monocationic chlorin (see Appendix A) with the known *Φ*_Δ_ value equaling 0.65 [14] was used as a standard PS. The DPBF destruction in both cuvettes was monitored spectrophotometrically at each minute, and each irradiation session contained 6–7 measurements. Then, the rate-of-degradation constants, *k*_d_, were evaluated in terms of the first-order exponential decay model. All the photobleaching experiments were repeated 5 to 6 times. The *Φ*_Δ_ values were computed as follows:(1)$$\Phi_{\Delta }=\frac{k_{d\;\mathrm{PS}}}{k_{d\;\mathrm{St}}}\;\frac{IP_{\mathrm{St}}}{IP_{\mathrm{PS}}}\Phi_{\Delta }{\;}_{\mathrm{St}},$$
where the St symbol refers to a solution of the monocationic chlorin. The *IP* values were estimated numerically by: (2)$$IP=\int_{\lambda_{1}}^{\lambda_{2}}I_{0}\left(\lambda \right){\left(1-10^{-A}\right)}_{\;}d\lambda ,$$
where *λ*_1_ and *λ*_2_ are the initial and final wavelengths of the region where the spectra of the LED panel and the PS studied overlap each other, *I*_0_ (*λ*) is the intensity of the panel between 590 and 720 nm as a function of *λ*, and *A* is the optical density of a solution.

The partition coefficients (*P*) between OctOH and phosphate saline buffer modeling the PS transfer from an aqueous to a lipid-like environment [27] were determined using the method of isothermal saturation [14,15], as follows:*P* = *m*_OctOH_/*m*_aq_,(3)
where *m*_OctOH_ and *m*_aq_ are the solute equilibrium molalities in OctOH and PSB, respectively. The equilibrium concentration of both PSs was analyzed spectrophotometrically. 

### 2.4. Administration and Activation of PSs In Vivo

“Fotoran e_6_” was usually administered via the intravenous route (see below). However, four allergic patients preferred to receive the PSs via the intravesical route. Before intravenous administration, both photosensitizers were dissolved in 200 mL of saline and then carefully administrated via slow infusion through an antecubital vein over 20–25 min at a dose of 0.7–1.4 mg/kg. During this procedure, each patient was monitored for adverse effects, such as drug hypersensitivity, tachycardia, or hypotension [7,8]. Between 2 and 3 h after PS infusion, TURB was performed followed by FD and PDT. For intravesical administration, the bladder was drained via an 18 Fr urethral catheter and then filled with 10 mL of “Fotoditazin” or 50 mg of “Fotoran e_6_” dissolved in 100 mL of saline. After 1.5–2 h, the bladder was drained again, and the manipulations mentioned above were initiated 1 h later. 

To activate PSs both for diagnostic (FD) and treatment (PDT) modes, a dual-channel, medical-grade ALXT-Elomed diode laser (Elomed, Russia) emitting at *λ* = 400 ± 1 nm or *λ* = 660 ± 1 nm was applied (see Appendix A). Before TURB was performed, the bladder was distended with saline to smooth the mucosal folds, corresponding to a bladder volume of 150 mL. Then rigid cystoscopic evaluation, with a continuous flow 22 Fr cystoscope, was carried out in white and then in blue light to detect the selective uptake of the PSs in tumor tissues. Further manipulations were very similar to those described earlier [4,7,8]. However, in the present study, we made an additional FD evaluation under blue light for detecting all suspicious foci of fluorescence to find residual flat tumors. As recommended elsewhere [4], the single PDT treatment was divided into two sessions. The first session consisted of irradiation of the tumor bed as well as the foci mentioned above using an optical fiber tool equipped with a microlens (Elomed, Russia), with the light dose of 150 J/cm^2^ delivered in 7–9 min. The second session was the diffuse irradiation of the entire mucosa with a 0.5 cm cylindrical diffuser located in the center of the bladder and the light dose of 10–25 J/cm^2^ delivered in 12–22 min. The light dose for this case was calculated in a similar way, as described elsewhere for a spherical bladder model [8,28] (see also the Appendix A). 

To assess the recurrence, control cystoscopy under white light was usually performed at 3-month intervals for 1 year. The surveillance interval was increased from here on to 6 months if recurrence was not detected. The patient was considered to be recurrence-free if cystoscopy in white light was normal [4,7,8].

## 3. Results and Discussion

### 3.1. Physicochemical Studies

Both chlorin PSs synthesized from natural chlorophyll have very similar chemical formulas [24]. However, an inspection of Figure 1 reveals two explicit differences in their structures. The first difference is that “Fotoditazin” is a dimeglumine salt of chlorin e_6_, whereas “Fotoran e_6_” is a trisodium salt. The second one is that “Fotoran e_6_” contains polyvinylpyrrolidone as an appropriate carrier. 

This PS formulation is believed to enhance the selectivity of PS accumulation in tumor tissues [24,28,29]. Despite the existence of a large number of clinical studies for both PSs (see [5,17,19,24] and references therein), experimental information about their behavior in liquids modeling the drug environment in vivo is scarce [28,29,30]. Below, we briefly consider some of the PS physicochemical characteristics that seem to be important for PDT efficacy.

The absorption spectra of both PSs are very similar and typical of chlorin macrocycles [24,31]. Figure 1 shows that they contain an intensive Soret (B-) band near 400 nm and less pronounced Q-bands between 500 and 670 nm. It is important that the most intensive Q_x(0-0)_-band at 660 nm, induced by π-π*-electron transfers, is within the so-called optical window of tissue [17,30,31]. The fluorescence spectra of PSs shown in Figure 1 demonstrate one band between 660 and 673 nm with a moderate Stokes shift. We have mentioned above that absorption of light by a PS molecule can transfer it to an excited triplet state. Then this molecule can interact with molecular oxygen leading to the formation of singlet oxygen, ^1^O_2_. The values of the ^1^O_2_ quantum yield (*Φ*_Δ_) given in Table 1 are seen to be rather high. This indicates that more than one-half of the excited PS molecules interact with molecular oxygen leading to the formation of ^1^O_2_, while the other ones transfer to their ground state mainly via fluorescence [17,23]. Hence, both PSs may be efficiently used for FD and PDT of various superficial tumors.

Table 1 also provides information about the PS partition in the OctOH/PSB biphasic system. We see that, for all the charged PSs, the corresponding partition coefficients are significantly smaller than those for neutral methylpheophorbide *a*, which is almost insoluble in an aqueous medium. Both “Fotoditazin” and “Fotoran e_6_” demonstrate moderate affinity toward the lipid-like compartment (*P*~2). These values are larger than those for the di- or tri-cationic chlorins (see Table 1) but are much smaller than the partition coefficient for the monocationic chlorin PS (*P*~9). This PS behavior is well-correlated with the fact that more hydrophobic MCh binds mainly to low-density lipoproteins in the vascular system, whereas anionic chlorin e_6_ is transported by serum albumin [32,33,34]. 

Thus, we see that both anionic PSs generate singlet oxygen with a sufficient quantum yield, demonstrate good absorption and fluorescence spectra for clinical utilization, and show a tendency to be localized in a lipid-like medium. However, their affinity toward a lipid-like compartment is rather weak (*P*~2). Hence, one may assume that such anionic PSs should be taken up by cancer cells mainly via endocytosis inducing photochemical damage in tumors via the Type II process of photochemical reactions and mainly targeting cytosol but not the mitochondria of malignant cells [5,17]. The application of a liposomal formulation for both PSs must enhance their affinity toward low-density lipoproteins and the selectivity of accumulation in malignant tissues.

### 3.2. Clinical Studies

We recruited 12 patients with NMIBC (see Table 2), including 6 males and 6 females, with a median age of 64.5 years (mean 61.3 ± 14.2, range 31–79). The patients underwent TURB, FD, and then a single PDT session. A total of 18 treatments were performed. Among the majority of the patients, the initial diagnosis was verified, excluding two cases where a biopsy of the resected tumors indicated a G2 histological grade. Additionally, in one patient, two suspicious foci were histologically identified as chronic inflammation (CI). Self-reported urinary adverse effects post TURB + FD + PDT usually persisted for one week and did not require any therapy (see Table 2). None of the patients developed phototoxicity or bladder contracture at either intravenous or intravesical PS administration.

The median follow-up was 24 months (mean 24.5 ± 5.4 months, range 16–35 months). One bedridden patient with several chronic diseases developed one histological recurrence without progression to the muscle layer. This patient underwent an additional TURB procedure. There were no recurrences in the other 11 patients during the follow-up period. Thus, TURB combined with intraoperative FD and PDT has reduced the recurrence rate of superficial bladder cancer to 9% for the one-year period of follow-up vs. 24–38% for the patients with T1 tumors who received TURB + CT [1] and 40–80% in the patients who received TURB as monotherapy [2]. This recurrence rate is also lower than the rate reported for the standard TURB + BCG treatment [1,2] and comparable with the results of TURB + PDT for the “Radachlorin” group [8]. It is worth noting that for the two-year period of follow-up, the recurrent rate in our study seems to be lower than the rate for the patients treated with “Radachlorin”. In our opinion, there are at least two reasons responsible for this difference. First, “Radachlorin” is a mixture of at least three chlorins compared to “Fotoditazin” and “Fotoran e_6_” which are individual compounds (see Figure 1). This fact may reduce the selectivity of PS accumulation in tumors. The second and most important is that the “Radachlorin” group consisted of patients with high-grade tumors refractory or intolerant to BSG. It is obvious that such patients reveal a more pronounced tendency for recurrence.

Table 3 compares the results of the PDT treatment of superficial bladder tumors with the second-generation PSs. We see that both aminolevulinic acid and its hexyl ether (HAL) are less efficient than chlorin PSs. In our opinion, this may result from the low rate of cellular uptake of ALA, the weak absorption of visible light by protoporphyrin IX, and the application of white incoherent light [35,36] for PS activation. In contrast, anionic chlorin PSs activated by coherent red light have been found to reduce the recurrence rate to 9% at the one-year follow-up period. Taking this fact into account, TURB + FD + PDT seems to be an attractive one-time option with good tolerability and minimal morbidity.

## 4. Conclusions

In conclusion, we can state the following as a result of the present and earlier studies using natural chlorin photosensitizers. (a) Our physicochemical results indicate that both “Fotoran e_6_” and “Fotoditazin” have good absorption of red light and fluorescence in the optical window of tissue. They generate singlet oxygen with a sufficient quantum yield and demonstrate moderate affinity toward a lipid-like compartment. (b) The pilot clinical study shows that TURB + FD + PDT with both PSs is a safe and tolerable modality for treating NMIBC. This technique seems to be highly efficient and of low toxicity compared to the standard treatment [1,4,5]. It is, however, apparent that further comparative trials with larger patient samples are strongly needed. (c) It is known that PDT can be successfully used in combination with chemotherapy or radiotherapy [17]. Hence, TURB + FD + PDT followed by BCG installations or intravesical chemotherapy must be an attractive approach for promoting the efficacy of the tumor treatment. In our opinion, several bladder instillations with a cytostatic of low toxicity, such as prospidium chloride [37], could provide an additional quality-of-life benefit for such patients. However, this extension of the research is left for future work.

## Figures and Tables

**Figure 1 jcm-11-00233-f001:**
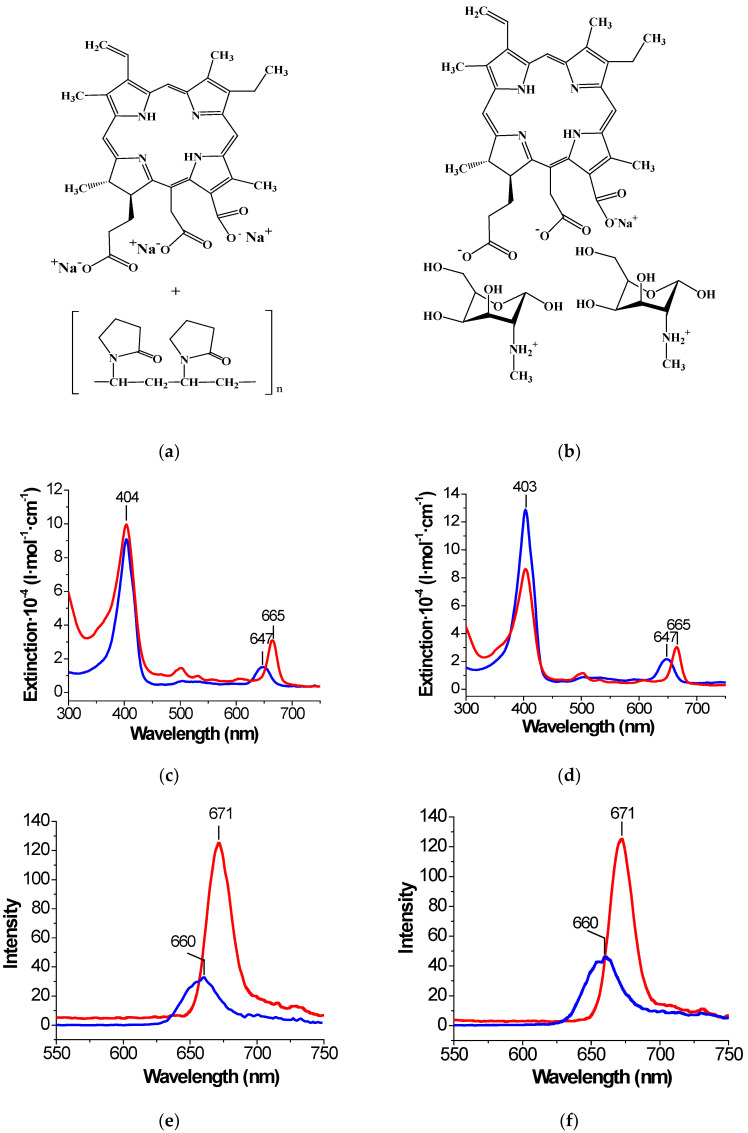
Molecular structures (**a**,**b**), absorption ((**c**,**d**) ~1 × 10^–5^ mol/kg), and fluorescence ((**e**,**f**); ~1 × 10^–6^ mol/kg) spectra of the PSs studied: (**a**,**c**,**e**)—“Fotoran e_6_” as a chlorin e_6_ trisodium salt; (**b**,**d**,**f**)—“Fotoditazin” as a dimeglumine sodium salt of chlorin e_6_. The blue lines show the results in water, while the red lines refer to liquid OctOH.

**Table 1 jcm-11-00233-t001:** Singlet oxygen quantum yield in OctOH and partition coefficients between OctOH and the phosphate saline buffer (pH = 7.4) for several chlorin PSs at 298 K.

	MCh ^1^	DCh ^1^	TCh ^1^	MPh *a* ^1^	Fotoran e_6_	Fotoditazin
*Φ* _Δ_	0.65 ± 0.07 ^2^	0.60 ± 0.06	0.53 ± 0.05	0.47 ± 0.05	0.56 ± 0.03	0.57 ± 0.02
*P*	8.6 ± 0.2	1.04 ± 0.02	0.97 ± 0.03	210.1 ± 6	1.88 ± 0.09	2.03 ± 0.21

^1^ The values are taken from refs. [14,22]. MPh *a*, MCh, DCh, and TCh denote methylpheophorbide *a*, mono-, di-, and tricationic chlorins, respectively. The PS structures are shown in Appendix A. ^2^ The uncertainties represent the twice standard deviation.

**Table 2 jcm-11-00233-t002:** Patient characteristics and TURB + FD + PDT outcomes ^1^.

N	Initials	Age	Sex	TN	TG	PS/Route	Follow-Up (Months)	Disease-Free Interval (Months)	AE/Remarks
1	PNF	66	M	1	G1	Fotoran e_6_/ Intravenous	16	16	Mild hematuria
2	EAV	79	F	1	G2	Fotoran e_6_/ Intravenous	19	1 h after 8 months (CIS, G2)	Mild hematuria/Underwent an additional TURB treatment
3	VNS	63	F	1 + 2(CI)	G2	Fotoran e_6_/ Intravenous	20	20	Mild hematuria, dysuria, frequent voiding
4	DNV	69	M	3	G1	Fotoran e_6_/ Intravenous	21	21	Mild hematuria, pain
5	KNN	67	M	1	G1	Fotoran e_6_/ Intravesical	24	24	Mild hematuria
6	DLV	74	F	1 + 1(CIS)	G1	Fotoran e_6_/ Intravesical	24	24	Mild hematuria
7	ASV	61	M	1	G1	Fotoran e_6_/ Intravenous	24	24	Dysuria
8	VMV	48	F	1	G2	Fotoran e_6_/ Intravenous	25	25	Mild pain
9	LNA	78	F	1	G2	Fotoran e_6_/ Intravenous	26	26	Dysuria, frequent voiding
10	KAS	49	M	1	G1	Fotoran e_6_/ Intravenous	27	27	Dysuria, frequent voiding
11	KPV	51	M	1	G1	Fotoditazin/ Intravesical	33	33	Dysuria, frequent voiding, low-grade fever for 1 day
12	LSV	31	F	1 + 1(CIS)	G1	Fotoditazin/ Intravesical	35	35	Mild hematuria, pain

^1^ TN, TG, and AE denote the tumor number, tumor grade, and adverse effects, respectively. Hr, CI, and CIS are histological recurrence, chronic inflammation, and carcinoma-in-situ. All the patients received a single TURB + FD + PDT treatment.

**Table 3 jcm-11-00233-t003:** Clinical studies for NMIBC treatment with second-generation PSs.

Authors	Year	Sample Size	PS	Light Dose/Route	Recurrence Rate	AE
Berger et al. [35]	2003	31	ALA	30–50 J/cm^2^/intravesical	48.4% (23.7 months)	Dysuria due to urinary tract infection, hematuria
Waidelich et al. [36]	2003	11	ALA	100 J/cm^2^ (incoherent white light)/intravesical	54.5% (18 months)	No systemic side-effects reported
Lee et al. [7]	2010	5	Fotolon	10 J/cm^2^/intravenous 20 J/cm^2^/intravesical	60% (29 months)	Dysuria, frequency, vesicoenteric fistula (one patient)
Bader et al. [6]	2013	17	HAL	25–100 J/cm^2^ (three treatments with incoherent white light)/intravesical	88% (21 months)	Transient bladder irritability, infection, gross hematuria
Lee et al. [8]	2013	34	Radachlorin	15 J/cm^2^/intravenous	9.1% (12 months) 39.9% (30 months)	Irritative bladder symptoms, infection, hematuria
Filonenko et al. [4]	2016	45	ALA	100 J/cm^2^ (tumor bed) 20 J/cm^2^ (all bladder)/intravesical	22% (12 months)	No complications reported
This work	2021	12	Fotoran e_6_/ Fotoditazin	150 J/cm^2^ (tumor bed) 10–25 J/cm^2^ (all bladder)/ intravenous or intravesical	8.3% (12 months) 8.3% ^1^ (24 months)	Mild hematuria, dysuria, frequent voiding, pain

^1^ Two patients treated in 2019 missed control cystoscopy during this summer. Our inspection of the Regional Oncological Dispensary service in October 2021 indicated that they were not treated there in 2021. Thus, these patients were nominally considered to be disease-free.

## Data Availability

The data presented in this study are available on request from the corresponding author.

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
