# Peer review of "Transurethral Resection of Non-Muscle Invasive Bladder Tumors Combined with Fluorescence Diagnosis and Photodynamic Therapy with Chlorin e6-Type Photosensitizers"

_jcm, 2021, doi:10.3390/jcm11010233_

Round 1
Reviewer 1 Report
TURB combined with FD and PDT are ineed a promising methods for the treatment of bladder cancer. However, the number of patients inrolled in this study is small, which can not explain this result, and the follow-up time is not long enough. It is recommended to submit after supplementing the number of cases.
Author Response
The Reviewer 1
TURB combined with FD and PDT are ineed a promising methods for the treatment of bladder cancer. However, the number of patients inrolled in this study is small, which can not explain this result, and the follow-up time is not long enough. It is recommended to submit after supplementing the number of cases.
Dear The Reviewer,
Thank You for the comment made. It is a great pity for us that the reviewer has not recommended this material for publication only due to a restricted number of patients. We agree that this point is important. However, as you can see from Table 3 this often occurs for the PDT treatment of bladder tumors. For example, two studies, viz. refs. 7, 36 included five and eleven patients, respectively. Moreover, You can see that our findings are in good agreement with those reported before for other chlorin PSs (see refs. 7, 8). Thus, considering these results all together, we are able to make at least a preliminary conclusion that TURB+FD+PDT is safe and efficient modality to treat superficial bladder tumors. We agree that additional studies with a larger sample size are required, however, in our opinion, the material reported here may be published in this special issue.
Sincerely yours
Andrey Kustov

Reviewer 2 Report
The authors report an interesting series of patients treated with two different chlorine photosensitizers administered by two different routes intravenous and installation at 2 concentrations. Based on all these conditions general conclusions can not be drawn and this should be a series of case studies.
It is not clear how the measured singlet oxygen quantum yield and the lipid partitioning coefficients influence the overall conclusions stemming from the clinical studies.
In the discussion the authors need to comment on the rather high fluence rate (~500 mW/cm^2) used and how this could affect the outcome, possibly, through thermal effects. It is unclear how the radiant exposure for focal illumination (150 J cm^-2) and systemic illumination of the bladder wall (10-25 J cm^-2) were determined. In the former case how was the illumination area determined and in the latter what is the variation across the bladder surface. Bladders are never spherical as in particular reproductive organs deform it, see DOI:10.1117/1.JBO.25.6.068001
Can the authors verify that the radiant exposure was in the 10-25 J cm^-2 range?
Figure 1 c and d can absorbance be modified into the molar extinction coefficient?
line 230 18 treatments were performed. It is unclear if multiple treatments were given in a single session or stretched out over multiple weeks/months? or are these the number of Identified tumours and 11 systematic illuminations?
Line 263 why coherent light? could any light be used.
Is there any information available about the chlorine PS concentration in the tumour samples, and normal bladder mucosa? Are there qualitative or subject differences in the PD contrasts between the two photosensitizer, route of administration or concentration? Is there equally an trend of correlation between the adverse events?
What conclusion regarding PD visualization and PDT did the authors draw to going forward with the one Photosensitizer, route of administration and PS concentration? If recurrent free survival is equal what is best as benefit for patients.
To evaluate the impact of the clinical outcome prior therapies and in particular immunomodulating therapies (BCG) did the 12 patient had. Does this impact survival as PDT is also an immune modulating therapy.
LIne 280. There is no data supporting the benefit of a low toxic cytostatic drug? Why is that here?
There is also some unnecessary information in this manuscript that can be removed.
- Line 58, 59 and 60 relating to antimicrobial PDT.
- Line 66 whereby no two photon studies would be applied in this concept of fiber-optical delivery with large surface areas.
Author Response
All comments with Figures are included in a separate file.

Reviewer 3 Report
This study in order to improve the treatment of bladder cancer, they evaluated the efficacy and safety of TURB+FD+PDT. Additionally,some crucial physicochemical parameters for both PSs were determined and discussed. Although the study is meaningful, several important issues need to be addressed prior to publication.
- In this study, only 24 patients were included ,Therefore, the conclusions drawn are less reliable. It is recommended to further expand the sample size to study.
- In the introduction, more description on mechanisms should be added.
- The conclusion part should be improved by a more comprehensive description.
- It is suggested to improve English expression abilities and check the manuscript carefully.
Author Response
The Reviewer 3
This study in order to improve the treatment of bladder cancer, they evaluated the efficacy and safety of TURB+FD+PDT. Additionally, some crucial physicochemical parameters for both PSs were determined and discussed. Although the study is meaningful, several important issues need to be addressed prior to publication.
Dear The Reviewer,
Thank You very much for your comments. Your questions and our answers are given below.
In this study, only 24 patients were included. Therefore, the conclusions drawn are less reliable. It is recommended to further expand the sample size to study.
Thank You for this important point. We agree that our study includes a small cohort of patients. However, you can see from Table 3 that this situation often takes place for the PDT treatment of bladder tumors. For example, two studies (see refs. 7, 36) included five and eleven patients, respectively. There are also two additional points which are worthy of note. Firstly, during last two years it has been very hard to recruit additional patients due to the COVID 19 pandemic and quarantine measures in the clinic. Secondly and finally, the results obtained are in good agreement with those reported before for other chlorin PSs (see refs. 7, 8). Thus, considering them all together, we are able to make at least a preliminary conclusion that TURB+FD+PDT is safe and efficient modality to treat superficial bladder tumors. We agree that additional studies with a larger sample size are required to confirm efficacy and safety of the treatment. Thus, we mentioned this point in the conclusion section and slightly changed the title of the paper.
In the introduction, more description on mechanisms should be added.
We modified this point.
The conclusion part should be improved by a more comprehensive description.
We totally modified conclusions.
It is suggested to improve English expression abilities and check the manuscript carefully.
We carefully checked and revised the manuscript. All the changes made were labeled by blue.
Sincerely yours
Andrey Kustov

Round 2
Reviewer 2 Report
The authors have responded well to the suggestions.
This reviewer would encourage to
Emphasis the fact that the patients had no prior BCG treatment. (Also note that BCG is misspelled in the conclusion section).
If you want to retain the antimicrobial section you need to make the link and the importance for Bladder cancer. The same holds true for the photophysical studies. Right now it is unclear how a physician would use the information for future clinical studies. You argued well in the rebuttal letter but this has not made it into the manuscript.
Author Response
Dear The Reviewer,
Thank you very much for your careful analysis of the revised version of the manuscript. We have modified the revised version according to your comments as full as possible. All modifications are labeled by green. Some of our comments are listed below.
The authors have responded well to the suggestions.
This reviewer would encourage to
Emphasis the fact that the patients had no prior BCG treatment. (Also note that BCG is misspelled in the conclusion section).
We have added this point.
If you want to retain the antimicrobial section you need to make the link and the importance for Bladder cancer. The same holds true for the photophysical studies. Right now it is unclear how a physician would use the information for future clinical studies. You argued well in the rebuttal letter but this has not made it into the manuscript.
You are right we want to retain both points. We have added several sentences into the main text for clarity. In our opinion, this is quite enough for this short paper.
Sincerely yours
Andrey Kustov

Reviewer 3 Report
accept
Author Response
The Reviewer 3
Moderate English changes are required.
Dear The Reviewer,
Thank You very much for your comment. We have carefully checked and revised the manuscript again. The new changes made were labeled by green.
Sincerely yours
Andrey Kustov
